# SCALING UP NEURAL ARCHITECTURE SEARCH WITH BIG SINGLE-STAGE MODELS

## ABSTRACT

Neural architecture search (NAS) methods have shown promising results discovering models that are both accurate and fast. For NAS, training a *one-shot model* has became a popular strategy to approximate the quality of multiple architectures (*child models*) using a single set of shared weights. To avoid performance degradation due to parameter sharing, most existing methods have a two-stage workflow where the best child model induced from the *one-shot model* has to be retrained or finetuned. In this work, we propose BigNAS, an approach that simplifies this workflow and scales up neural architecture search to target a wide range of model sizes simultaneously. We propose several techniques to bridge the gap between the distinct initialization and learning dynamics across small and big models with shared parameters, which enable us to train a *single-stage model*: a single model from which we can directly slice high-quality child models *without retraining or finetuning*. With BigNAS we are able to train a single set of shared weights on ImageNet and use these weights to obtain child models whose sizes range from 200 to 1000 MFLOPs. Our discovered model family, BigNASModels, achieve top-1 accuracies ranging from 76.5% to 80.9%, surpassing all state-of-the-art models in this range including EfficientNets.

## 1 INTRODUCTION

Designing network architectures that are both accurate and efficient is crucial for deep learning on edge devices. It is well known that a single neural network architecture can require more than an order of magnitude more inference time if it is deployed on a slower device (Yu et al., 2018). This makes it appealing to not only search for architectures that are optimized for specific devices, but also to ensure that a range of models can be deployed effectively.

In the past, Neural Architecture Search (NAS) methods (Zoph & Le, 2016; Zoph et al., 2018; Real et al., 2018) have shown to be excellent at optimizing for a single device and latency target (Tan et al., 2019). However, if we wish to target a large array of devices, it becomes prohibitively expensive and time-consuming to run a separate search for each one. A possible solution to this is to use basic scaling heuristics such as the EfficientNet family (Tan & Le, 2019). However, this loses out on opportunities to optimize models specialized for the diverse performance characteristics of individual devices. Another option would be to use efficient architecture search methods, *e.g.*, Pham et al. (2018); Bender et al. (2018); Liu et al. (2018b). However, to target multiple devices, we must run many searches and retrain all of the searched models from scratch.

In this work, we search over a big *single-stage* model that contains both small child models (∼200 MFLOPs, comparable to MobileNetV3) and big child models (∼1 GFLOPs, comparable to EfficientNet B2). Different from existing one-shot methods (Bender et al., 2018; Liu et al., 2018b; Brock et al., 2018; Pham et al., 2018), our trained single-stage model offers a much wider coverage of model capacities, and more importantly, all child models are trained in a way such that they simultaneously reach excellent performance at the end of the search phase, without requiring a separate retraining step. Architecture selection can be then carried out via a simple coarse-to-fine selection strategy. Once an architecture is selected, we can obtain a child model by slicing the single-stage model for instant deployment w.r.t. the given constraints such as memory footprint and/or runtime latency. The workflow is illustrated in Figure 1.

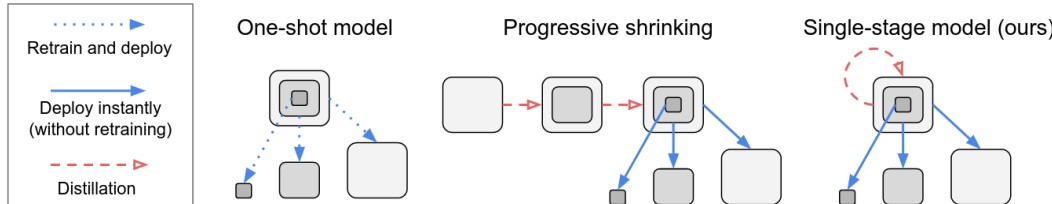

Figure 1: Comparison with several existing workflows. We use nested squares to denote models with shared weights, and use the size of the square to denote the size of each model. Workflow in the middle refers the concurrent work from Cai et al. (2019), where submodels are sequentially induced through progressive distillation and channel sorting. We simultaneously train all child models in a single-stage model with proposed modifications, and deploy them without retraining or finetuning.

The success of our method heavily relies on training a high-quality single-stage model, which is challenging on its own. For example, we find the training loss explodes if the single-stage model is not properly initialized, and bigger child models start to overfit before smaller ones plateau. It is particularly nontrivial to simultaneously retain good performance on every individual child model due to aggressive parameter sharing during architecture search. We address these challenges through a combination of techniques, including an improved sampling strategy and efficient inplace distillation, and substantially stabilize the single-stage model training through better initialization, learning rate schedule and regularization. Effectiveness of the proposed solutions is backed up by ablation studies.

With the proposed techniques, we are able train a single-stage model on ImageNet and obtain a family of child models that simultaneously surpass all the state-of-the-art models in the range of 200 to 1000 MFLOPs, including EfficientNets B0-B2 (1.6% more accurate under 400 MFLOPs), without retraining or finetuning the child models upon the completion of search. One of our child models achieves 80.9% top-1 accuracy at 1G FLOPs (four times less computation than a ResNet-50).

## 2 RELATED WORK

Earlier NAS methods (Zoph & Le, 2016; Zoph et al., 2018; Liu et al., 2017; 2018a; Real et al., 2018) train thousands of candidate architectures from scratch (on a smaller proxy task) and use their validation performance as the feedback to an algorithm that learns to focus on the most promising regions in the search space. More recent works have sought to amortize the cost by training a single over-parameterized *one-shot model*. Each architecture in the search space uses only a subset of the operations in the one-shot model; these *child models* can be efficiently ranked by using the shared weights to estimate their relative accuracies (Brock et al., 2018; Pham et al., 2018; Bender et al., 2018; Liu et al., 2018b; Cai et al., 2018; Wu et al., 2019).

As a complementary direction, resource-aware NAS methods are proposed to simultaneously maximize prediction accuracy and minimize resource requirements such as latency, FLOPs, or memory footprints (Tan et al., 2019; Cai et al., 2019; Wu et al., 2019; Stamoulis et al., 2019; Guo et al., 2019; Yu & Huang, 2019a).

All the aforementioned approaches require two-stage training: Once the best architectures have been identified (either through the proxy tasks or using a one-shot model), they have to be retrained from scratch to obtain a final model with higher accuracy. In most of these existing works, a single search experiment only targets a single resource budget or a narrow range of resource budgets at a time.

To alleviate these issues, Cai et al. (2019) proposed a progressive training approach (OFA) concurrently with our work. The idea is to pre-train a single full network and then progressively distill it to obtain the smaller networks. Moreover, a channel sorting procedure is required to progressively construct the smaller networks. In our proposed BigNAS, however, all the child models in the single-stage model are trained *simultaneously*, allowing the learning of small and big networks to mutually benefit each other. During the training, we always keep lower-index channels in each layer and lower-index layers in each stage for our child models, eliminating the sorting procedure. While OFA focuses on a limited range of model sizes (for example, models around 300 MFLOPs),

our BigNAS is able to handle a wider set of models (from 200 MFLOPs to 1 GFLOPs) and offers a better coverage over diverse deployment scenarios and varied resource budgets.

Our work shares high-level similarities with *slimmable networks* (Yu et al., 2018; Yu & Huang, 2019b;a) in terms of training a single shared set of weights which can be used for many child models. However, while slimmable networks are specialized to vary the number of channels only, we are able to handle a much larger space where many architectural dimensions (kernel and channel sizes, network depths, input resolutions) are searched simultaneously, subsuming and outperforming the scaling heuristics in EfficientNets (Tan & Le, 2019).

## 3   ARCHITECTURE SEARCH WITH SINGLE-STAGE MODELS

Our proposed method consists of two steps:

1. We train a big *single-stage model* from which we can directly sample or slice different architectures as *child models* for instant inference and deployment. In contrast to previous works (Brock et al., 2018; Pham et al., 2018; Bender et al., 2018; Liu et al., 2018b; Stamoulis et al., 2019; Guo et al., 2019), our training is single-stage as it does not require finetuning the sampled architectures or retraining them from scratch at the end of search.
2. Architecture selection using a simple *coarse-to-fine* selection method to find the most accurate model under the given resource constraints (for example, FLOPs, memory footprint and/or runtime latency budgets on different devices).

In the following, we will first systematically study how to train a *high-quality single-stage model* from five aspects: network sampling during training, inplace distillation, network initialization, convergence behavior and regularization. Then we will present a coarse-to-fine approach for efficient resource-aware architecture selection.

### 3.1   TRAINING A HIGH-QUALITY SINGLE-STAGE MODEL

Training a high-quality single-stage model is important and highly non-trivial due to the distinct initialization and learning dynamics of small and big child models. In this section, we first generalize two techniques originally introduced by Yu & Huang (2019b) to simultaneously train a set of high-quality networks with different channel numbers, and show that both can be extended to handle a much larger space where the architectural dimensions, including kernel sizes, channel numbers, input resolutions, network depths are jointly searched. We then present three additional techniques to address the distinct initialization and learning dynamics of small and big child models.

**Sandwich Rule**   In each training step, given a mini-batch of data, the sandwich rule (Yu & Huang, 2019b) samples the smallest child model, the biggest (full) child model and $N$ randomly sampled child models ($N = 2$ in our experiments). It then aggregates the gradients from all sampled child models before updating the weights of the single-stage model. As multiple architectural dimensions are included in our search space, the "smallest" child model is the one with lowest input resolution, thinnest width, shallowest depth, and smallest kernel size (the kernel of the depthwise convolutions in each inverted residual block (Sandler et al., 2018)). The motivation is to improve all child models in our search space simultaneously, by pushing up both the performance lower bound (the smallest child model) and the performance upper bound (the biggest child model) across all child models.

**Inplace Distillation**   During the training of a single-stage model, inplace distillation (Yu & Huang, 2019b) takes the soft labels predicted by the biggest possible child model (full model) to supervise all other child models. The benefit of inplace distillation comes for free in our training setting, as we always have access to the predictions of the largest child model in each gradient update step thanks to the sandwich rule. We note that all child models are only trained with the inplace distillation loss, starting from the first training step to the end of the training.

During training, input images are randomly cropped as a preliminary data augmentation step. When distilling a high-resolution teacher model into a low-resolution student model, we find that it is helpful to feed the same image patches into both the teacher and the student. In our data preparation, we first randomly crop an image with a fixed resolution (on ImageNet we use 224), and then apply

bicubic interpolation to the *same patch* to transform it into all target resolutions (*e.g.*, 192, 288, 320). In this case, soft labels predicted by the biggest child model (the teacher) are more compatible with the inputs seen by other child models (the students). Therefore this can serve as a more accurate distillation signal. Our preliminary results show that sampling different patches even in a same image leads to $\sim 0.3\%$ drop on top-1 accuracy for child models.

**Initialization** Previous weight initialization methods, such as "He Initialization" (He et al., 2015), are deduced from fixed neural networks where the number of input units $n$ (the fan-in) is constant. The principal motivation of these initialization methods is to keep the variance of the responses in each layer unchanged, so that the forward information signals and the backward gradients will not be reduced or magnified exponentially as the network goes deeper. For example, He et al. (2015) suggested to initialize the variance of the weights as $\frac{2}{n}$ for convolutions with ReLU activations.

However, the above is ill-fitted for initializing a *single-stage model*, where $n$ is no longer a constant across different child models with varied kernel sizes and input channels. This issue is exaggerated when we train bigger and deeper single-stage models. In practice, we find the training loss of a single-stage model explodes when we use the optimized learning rates for training a normal network. The training starts to work when we reduce the learning rate to 30%, but it leads to much worse results ($\sim 1.0\%$ top-1 accuracy on ImageNet).

Identifying this issue is critical while the solution is quite simple. As all child models in our search space are residual networks, we initialize the output of each residual block (before skip connection) to an all-zeros tensor by setting the learnable scaling coefficient $\gamma = 0$ in the last Batch Normalization (Ioffe & Szegedy, 2015) layer of each residual block, ensuring identical variance before and after each residual block regardless of the fan-in. This initialization is originally mentioned in (Goyal et al., 2017) which improves accuracy by $\sim 0.2\%$ in their setting, yet is more critical in our setting (improving by $\sim 1.0\%$) due to the above analyzed initialization issue. We also additionally add a skip connection between each stage transitions when either resolutions or channels differ (using $2 \times 2$ average pooling and/or $1 \times 1$ convolution if necessary) to explicitly construct an identity mapping (He et al., 2016b).

**Convergence Behavior** In practice, we find that big child models converge faster while small child models converge slower. Figure 2a shows the typical learning curves during the training of a single-stage model, where we plot the validation accuracies of a small and a big child model over time. This reveals a dilemma: at training step $t$ when the performance of big child models peaks, the small child models are not fully-trained; and at training step $t'$ when the small child models have better performance, the big child models already overfitted.

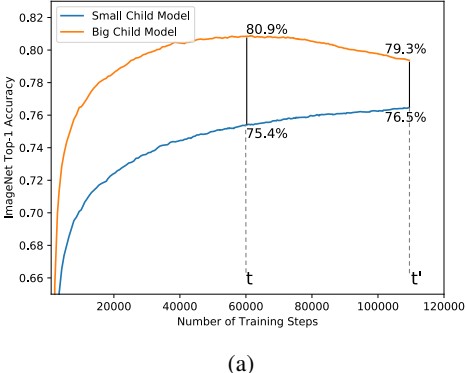
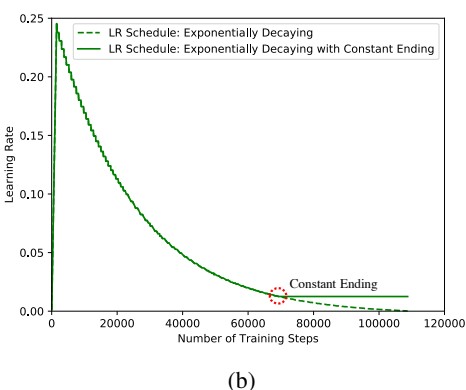

(a)                                              (b)

Figure 2: On the left, we show typical accuracy curves during the training process for both small and big child models. On the right, we plot the modified learning rate schedules with constant ending.

To address this issue, we put our focus on the learning rate schedule. We first plot the optimized and widely used exponentially decaying learning rate schedule for MobileNet-series (Howard et al., 2017; Sandler et al., 2018; Howard et al., 2019), MNasNets (Tan et al., 2019) and EfficientNets (Tan & Le, 2019) in Figure 2b. We introduce a simple modification to this learning rate schedule, named

*exponentially decaying with constant ending*, which has a constant learning rate at the end of training when it reaches 5% of the initial learning rate (Figure 2b). It brings two benefits. First, with a slightly larger learning rate at the end, the small child models learn faster. Second, the constant learning rate at the end alleviates the overfitting of big child models as the weights oscillate.

**Regularization**  Big child models tend to overfit the training data whereas small child models tend to underfit. In previous work, Bender et al. (2018) apply the same weight decay to all child models regardless whether they are small or big. In EfficientNet, Tan & Le (2019) linearly increase dropout (Srivastava et al., 2014) ratio as it moves from the smallest EfficientNet-B0 to the biggest EfficientNet-B7. This becomes even more complicated in the context of training big single-stage models, due to the interplay among the small child models and big child models with shared parameters. Nevertheless, we introduce a simple rule that is surprisingly effective for this problem: *regularize only the biggest (full) child model* (*i.e.*, the only model that has direct access to the ground truth training labels because of inplace distillation). We apply this rule to both weight decay and dropout, and empirically demonstrate its effectiveness in our experiments.

**Batch Norm Calibration**  Batch norm statistics are not accumulated when training the single-stage model as they are ill-defined with varying architectures. After the training is completed, we re-calibrate the batch norm statistics (Yu & Huang, 2019b) for each sampled child model for deployment without retraining or finetuning any network parameters.

## 3.2  Coarse-to-fine Architecture Selection

After training a single-stage model, one needs to select the best architectures w.r.t. the resource budgets. Although obtaining the accuracy of a child model is cheap, the number of architecture candidates is extremely large (more than $10^{12}$). To address this issue, we propose a coarse-to-fine strategy where we first try to find a rough skeleton of promising network candidates in general, and then sample multiple fine-grained variations around each skeleton architecture of interest.

Specifically, in the coarse-grained phase, we define a limited input resolution set, depth set (global depth multipliers), channel set (global width multipliers) and kernel size set, and obtain benchmarks for all child models in this restricted space. This is followed by a fine-grained search phase, where we first pick the best network skeleton satisfying the given resource constraint found in the previous phase, and then randomly mutate its network-wise resolution, stage-wise depth, number of channels and kernel sizes to further discover better network architectures. Finally, we directly use the weights from the single-stage model for the induced child models without any retraining or finetuning. More details will be presented in the experiments.

## 4  Experiments

In this section, we first present the details of our search space, followed by our main results compared with the previous state-of-the-arts in terms of both accuracy and efficiency. Then we conduct extensive ablative study to demonstrate the effectiveness of our proposed modifications. Finally, we show the intermediate results of our coarse-to-fine architecture selection.

## 4.1  Search Space Definition

Following previous resource-aware NAS methods (Tan et al., 2019; Tan & Le, 2019; Cai et al., 2018; Wu et al., 2019; Howard et al., 2019; Wu et al., 2019), our network architectures consist of a stack with inverted bottleneck residual blocks (MBConv) (Sandler et al., 2018). The detailed search space is summarized in Table 1. For the input resolution dimension, we sample from set {192, 224, 288, 320}. By training on different input resolutions, we find our trained single-stage model is able to generalize to unseen input resolutions during architecture search or deployment (*e.g.*, 208, 240, 256, 272, 304, 336) after BN calibration. For the depth dimension, our network has seven stages (excluding the first and the last convolution layer). Each stage has multiple choices of the number of layers (*e.g.*, stage 5 can pick any number of layers ranging from 2 to 6). Following slimmable networks (Yu et al., 2018) that always keep lower-index channels in each layer, we always keep *lower-index layers* in each network stage (and their weights). For weight sharing on the kernel size

Table 1: MobileNetV2-based search space.

| Stage | Operator | Resolution | #Channels | #Layers | Kernel Sizes |
|---|---|---|---|---|---|
| | Conv | $192 \times 192$ - $320 \times 320$ | 32 - 40 | 1 | 3 |
| 1 | MBConv1 | $96 \times 96$ - $160 \times 160$ | 16 - 24 | 1 - 2 | 3 |
| 2 | MBConv6 | $96 \times 96$ - $160 \times 160$ | 24 - 32 | 2 - 3 | 3 |
| 3 | MBConv6 | $48 \times 48$ - $80 \times 80$ | 40 - 48 | 2 - 3 | 3, 5 |
| 4 | MBConv6 | $24 \times 24$ - $40 \times 40$ | 80 - 88 | 2 - 4 | 3, 5 |
| 5 | MBConv6 | $12 \times 12$ - $20 \times 20$ | 112 - 128 | 2 - 6 | 3, 5 |
| 6 | MBConv6 | $12 \times 12$ - $20 \times 20$ | 192 - 216 | 2 - 6 | 3, 5 |
| 7 | MBConv6 | $6 \times 6$ - $10 \times 10$ | 320 - 352 | 1 - 2 | 3, 5 |
| | Conv | $6 \times 6$ - $10 \times 10$ | 1280 - 1408 | 1 | 1 |

dimension in the inverted residual blocks, a $3 \times 3$ depthwise kernel is defined to be the center of a $5 \times 5$ depthwise kernel. Both kernel sizes and channel numbers can be adjusted layer-wise. The input resolution is network-wise and the number of layers is a stage-wise configuration in our search space.

## 4.2 MAIN RESULTS ON IMAGENET

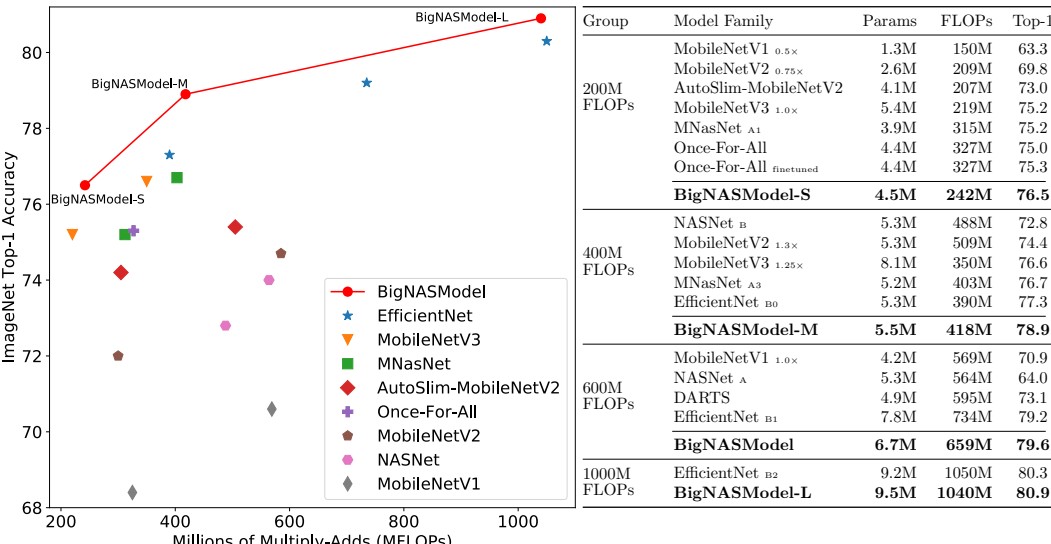

Figure 3: Main results of BigNASModels on ImageNet.

We train our big single-stage model on ImageNet (Deng et al., 2009) using same settings as (Tan et al., 2019; Tan & Le, 2019; Howard et al., 2017). We use RMSProp optimizer with decay 0.9 and momentum 0.9; batch normalization with post-calibration (Yu & Huang, 2019b); weight decaying factor $1e-5$; initial learning rate 0.256 that decays by 0.97 every 2.4 epochs. We also use the swish activation (Ramachandran et al., 2017) and fixed AutoAugment V0 policy (Cubuk et al., 2018) following EfficientNets (Tan & Le, 2019). In addition to these training settings, we train our big single-stage model with all techniques proposed in Section 3.1. The learning rate is truncated to a constant value when it reaches 5% of its initial learning rate (*i.e.*, 0.0128) until the training ends. We apply dropout only on training the full network with dropout ratio 0.2, and weight decaying only on full network once in each training iteration. To train the single-stage model, we adopt the sandwich sampling rules and inplace distillation proposed by Yu & Huang (2019b). After the training, we

use a simple coarse-to-fine architecture selection to find the best architecture under each interested resource budgets. We will show the details of coarse-to-fine architecture selection in Section 4.4.

We show the performance benchmark of our model family, named BigNASModels, in Figure 3. On the left we show the visualization of FLOPs-Accuracy benchmarks compared with the previous state-of-the-arts including MobileNetV1 (Howard et al., 2017), NASNet (Zoph et al., 2018), MobileNetV2 (Sandler et al., 2018), AutoSlim-MobileNetV2 (Yu & Huang, 2019a), MNasNet (Tan et al., 2019), MobileNetV3 (Howard et al., 2019), EfficientNet (Tan & Le, 2019) and concurrent work Once-For-All (Cai et al., 2019). We show the detailed benchmark results on the right table. For small-sized models, our BigNASModel-S achieves 76.5% accuracy under only 240 MFLOPs, which is 1.3% better than MobileNetV3 in terms of similar FLOPs, and 0.5% better than ResNet-50 (He et al., 2016a) with 17 × fewer FLOPs. For medium-sized models, our BigNASModel-M achieves 1.6% better accuracy than EfficientNet B0. For large-sized models, even when ImageNet classification accuracy saturates, our BigNASModel-L still has 0.6% improvement compared with Efficient-Net B2. Moreover, instead of individually training models of different sizes, our BigNASModel-S, BigNASModel-M and BigNASModel-L are sliced directly from one pretrained single-stage model, without retraining or finetuning.

## 4.3 ABLATION STUDY

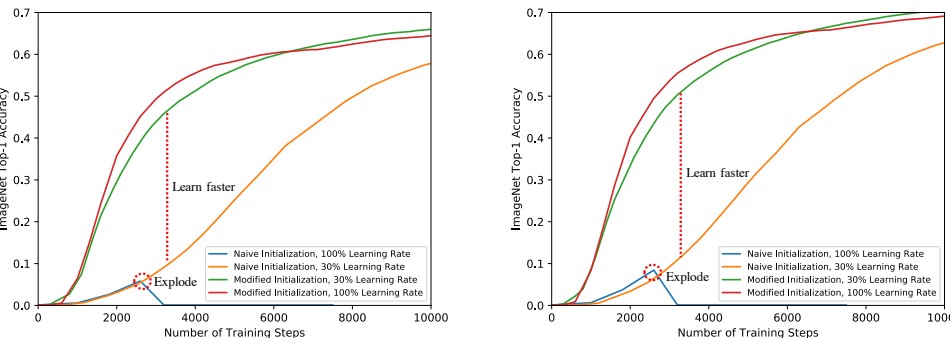

Figure 4: **Focusing on the start of training.** Ablation study on different initialization methods. We show the validation accuracy of a small (left) and big (right) child model.

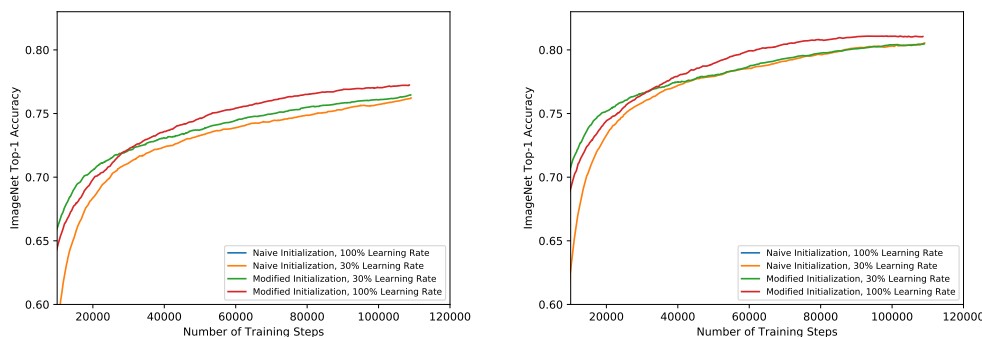

Figure 5: **Focusing on the end of training.** Ablation study on different initialization methods. We show the validation accuracy of a small (left) and big (right) child model.

**Ablation Study on Initialization.** Previous weight initialization methods (He et al., 2015) are deduced from fixed neural networks, where the numbers of input units is constant. However, in a single-stage model, the number of input units varies across the different child models. In this part, we start with training a single-stage model using He Initialization (He et al., 2015) designed for fixed neural networks. As shown in Figure 4, the accuracy of both small (left) and big (right) child

models drops to zero after a few thousand training steps during the learning rate warming-up (Goyal et al., 2017). The single-stage model is able to converge when we reduce the learning rate to the 30% of its original value. If the initialization is modified according to Section 3.1, the model learns much faster at the beginning of the training (shown in Figure 4), and has better performance at the end of the training (shown in Figure 5). Moreover, we are also able to train the single-stage model with the original learning rate hyper-parameter, which leads to much better performance for both small (Figure 5, left) and big (Figure 5, right) child models.

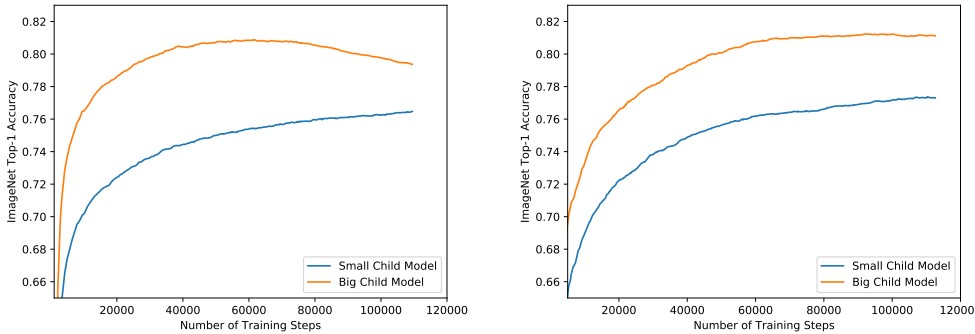

Figure 6: The validation accuracy curves during the training process for both small and big child models before (left) and after (right) our modifications.

**Ablation Study on Convergence Behavior.** During the training of a single-stage model, the big child models converge faster and then overfit, while small child models converge slower and need more training. In this part, we show the performance after addressing this issue in Figure 6. We apply the proposed learning rate schedule *exponentially decaying with constant ending* on the right. The detailed learning rate schedules are shown in Figure 2b. We also tried many other learning rate schedules with an exhaustive hyper-parameter sweep, including linearly decaying (Ma et al., 2018; Yu & Huang, 2019b) and cosine decaying (Loshchilov & Hutter, 2016; He et al., 2019). But the performances are all worse than exponentially decaying.

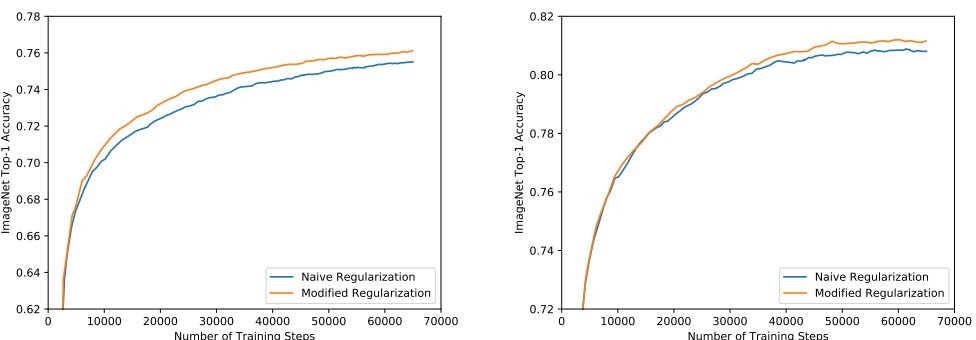

Figure 7: The validation accuracy of a small (left) and big (right) child model using different regularization rules.

**Ablation Study on Regularization.** Big child models are prone to overfitting on the training data whereas small child models are prone to underfitting. In this part, we compare the effects of the regularization between two rules: (1) applying regularization on all child models (Bender et al., 2018), and (2) applying regularization only on the full network. Here the regularization techniques we consider are weight decay with factor $1e-5$ and dropout with ratio $0.2$ (the same hyper-parameters used in training previous state-of-the-art mobile networks). In Figure 7, we show the performance of both small (left) and big (right) child models using different regularization rules. On the left, the performance of small child models is improved by a large margin ($+0.5$ top-1 accuracy) as it has

less regularization and more capacity to fit the training data. Meanwhile on the right, we found the performance of the big child model is also improved slightly (+0.2 top-1 accuracy).

### 4.4 COARSE-TO-FINE ARCHITECTURE SELECTION

After the training of a single-stage model, we use coarse-to-fine architecture selection to find the best architectures under different resource budgets. During the search, the evaluation metrics can be flexible including predictive accuracy, FLOPs, memory footprint, latency on various different devices, and many others. It is noteworthy that we pick the best architectures according to the predictive accuracy on training set, because we used all training data for obtaining our single-stage model (no retraining from scratch), and the validation set of ImageNet (Deng et al., 2009) is being used as "test set" in the community. In this part, we first show an illustration of our coarse-to-fine architecture selection with the trained big single-stage model in Figure 8. The search results are based on FLOPs-Accuracy benchmarks (as FLOPs are more reproducible and independent of the software version, hardware version, runtime environments and many other factors).

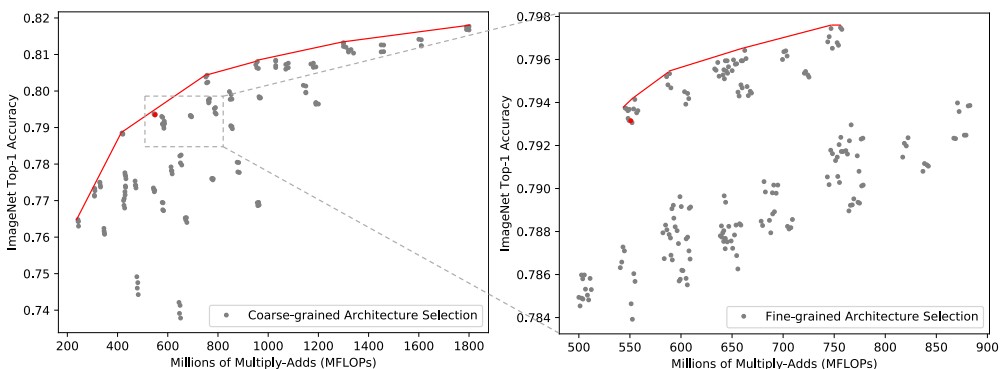

Figure 8: Benchmark results of coarse-to-fine architecture selection. The red dot in coarse-grained architecture selection is picked and mutated for fine-grained architecture selection.

During the coarse-to-fine architecture selection, we first find rough skeletons of good candidate networks. Specifically, in the coarse selection phase, we pre-define five input resolutions (network-wise, {192, 224, 256, 288, 320}), four depth configurations (stage-wise via global depth multipliers (Tan & Le, 2019)), two channel configurations (stage-wise via global width multipliers (Howard et al., 2017)) and four kernel size configurations (stage-wise), and obtain all of their benchmarks (shown in Figure 8 on the left). Then under our interested latency budget, we perform a fine-grained grid search by varying its configurations (shown in Figure 8 on the right). For example, under FLOPs near 600M we first pick the skeleton of the red dot shown in Figure 8. We then perform additional fine-grained architecture selection by randomly varying the input resolutions, depths, channels and kernel sizes slightly. We note that the coarse-to-fine architecture selection is flexible and not very exhaustive in our experiments, yet it already discovered fairly good architectures as shown in Figure 8 on the right. For the FLOPs near 650M, we finally select the child model with input resolution 256, depth configuration {1:2:2:2:4:4:1}, channel configuration {32:16:24:48:88:128:216:352:1408} and kernel size configuration {3:3:5:3:5:5:3}. After training of the single-stage model, the post-search step is highly parallelizable and independent of training.

## 5 CONCLUSION

We presented a novel paradigm for neural architecture search by training a single-stage model, from which high-quality child models of different sizes can be induced for instant deployment without retraining or finetuning. With several proposed techniques, we obtain a family of BigNASModels as slices in a big pre-trained single-stage model. These slices simultaneously surpass all state-of-the-art ImageNet classification models ranging from 200 MFLOPs to 1G FLOPs. We hope our work can serve to further simplify and scale up neural architecture search.

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

## A    Architectures of BigNASModel

We show the architecture visualization of the single-stage model and child models BigNASModel-S, BigNASModel-M, BigNASModel@660M, BigNASModel-L in Figure 9. The child models are directly sliced from the single-stage model without retraining or finetuning. Compared with the compound model scaling heuristic (Tan & Le, 2019), our child models have distinct architectures across all dimensions. For example, comparing BigNASModel-L with EfficientNet-B2, the EfficientNet-B2 has input resolution 260, channels $\{40{:}24{:}32{:}40{:}88{:}128{:}216{:}352{:}1408\}$, kernel sizes $\{3{:}3{:}5{:}3{:}5{:}5{:}3\}$ and stage layers $\{2{:}3{:}3{:}4{:}4{:}5{:}2\}$. Our BigNASModel-L achieves 80.9% top-1 accuracy under 1040 MFLOPs, while EfficientNet-B2 achieves 80.3% top-1 accuracy under 1050 MFLOPs.

## B    Implementation Details

We implement all training and coarse-to-fine architecture selection algorithms on TensorFlow framework (Abadi et al., 2015). All of our experiments are conducted on $8 \times 8$ TPUv3 pods. For ImageNet experiments, we use a total batch size 4096. Our single-stage model has sizes from 200 to 2000 MFLOPs, from which we search architectures from 200 to 1000 MFLOPs. To train a single-stage model, it roughly takes 36 hours.

Training on TPUs requires defining a static computational graph, where the shapes of all tensors in that graph should be fixed. Thus, during the training we are not able to dynamically slice the weights, select computational paths or sample many input resolutions. To this end, here we provide the details of our implementation for training single-stage models on TPUs. On the dimensions of kernel sizes, channels, and depths, we use the masking strategy to simulate the weight slicing or path selection during the training (*i.e.*, we mask out the rest of the channels, kernel paddings, or the entire output of a residual block). On the dimension of input resolutions, in each training iteration, our data pipeline provides same images with four fixed resolutions ($\{192, 224, 288, 320\}$) which are paired with the model sizes. The smallest child model is always trained on the lowest resolution, while the biggest child model is always trained on the highest resolution. For all other resolutions the models are randomly varied on kernel sizes, channels, and depths. By this implementation, our trained single-stage model is able to provide high-quality child models across all these dimensions. For inference, we directly declare a child model architecture and load the sliced weights from the single-stage model. To slice the weights, we always use lower-index channels in each layer, lower-index layers in each stage, and the center $3 \times 3$ depthwise kernel from a $5 \times 5$ depthwise kernel.

For the data prefetching pipeline, we need multiple image input resolutions during the training. We first prefetch a batch of training patches with a fixed resolution (on ImageNet we use 224) with data augmentations, and then resize them with bicubic interpolation to our target input resolutions (*e.g.*, 192, 224, 288, 320). We note that during inference, the single-crop testing accuracy is reported. Importantly, for testing data prefetching pipeline, we also prefetch a 224 center crop first and then resize to the target resolution to avoid the inconsistency.

During the training, we use cross-replica (synchronized) batch normalization following Efficient-Nets (Tan & Le, 2019). To enable this, we also have to use stateless random sampling function [1] since naive random sampling function [2] leads to different sampled values across different TPU cores. The input seed of stateless random sampling functions is the global training step plus current layer index so that the trained single-stage model can provide child models with different layer-wise/stage-wise configurations.

---

[1] https://www.tensorflow.org/api_docs/python/tf/random/stateless_uniform
[2] https://www.tensorflow.org/api_docs/python/tf/random/uniform

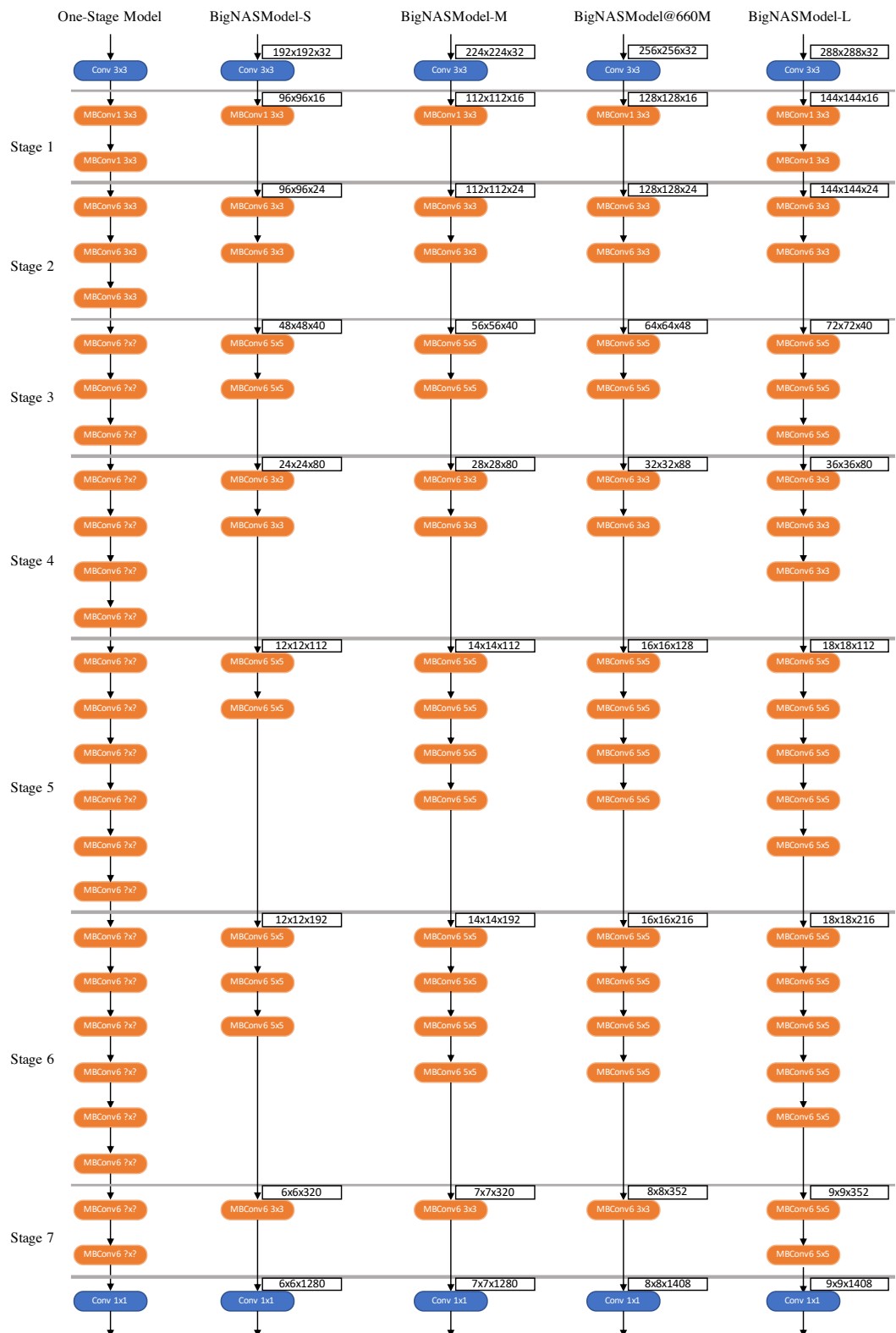

Figure 9: Architecture visualization of the single-stage model and child models BigNASModel-S, BigNASModel-M, BigNASModel@660M, BigNASModel-L. All child models are directly sliced from the single-stage model without retraining or finetuning.

