# OpenReview forum: "Scaling Up Neural Architecture Search with Big Single-Stage Models"
_ICLR.cc/2020/Conference — Reject_

### Official Review · AnonReviewer1 · 2019-10-22
**Official Blind Review #1**

**Rating:** 3

**Review:**

Parameter sharing (PS) is an important approach to speed up neural network search (NAS), which further allows the development of differential architecture search (e.g., DARTS) methods. However, PS also deteriorates the performance of learning models on the validation/testing set.

This paper first changes the search space from a DAG (micro+marco) in e.g., DARTS to a stacked one based on MBConv; and then, propose to use several tricks to train the super-net well. Finally, a search method is constructed for the supernet to find the desired architectures.

Overall, the paper is too experimental. The method is an ensemble of existing approaches, i.e., every single component in the paper has been visited in the literature. Expect for experimental results, I do not see many general lessons we can learn from the paper. Finally, why the proposed method can be better than others is not well-explained and clarified.

Please see the questions below:

Q1. Is NAS a method only for ImageNet? Can the method generalize to more applications/datasets?
- While ImageNet is a good dataset for CV experiments, I think NAS should be a method for deriving architectures with certain requirements.
- So, with so many tricks proposed in the paper, I wish authors can carry on experiments on other data sets as well, e.g., CIFAR and MNIST, which can still be preferred.

Q2. On motivation, could authors explain more about the difficulties of combining all these techniques?
- Each method is brought from some other paper, what motivate authors to combine them together? What makes them believe this is possible?

Q3. On presentation, could authors draw a figure of the search space in the main text and give an overall algorithm for Section "3.2 COARSE-TO-FINE ARCHITECTURE SELECTION". It is hard for a reader to see novelties there.

Q4. "We also use the swish activation (Ramachandran et al., 2017) and fixed AutoAugment V0 policy" - are all other compared methods using swish activation and AutoAugment V0 policy?

Q5. How about the search efficiency of the proposed method? Only the accuracy is reported in the paper.

Q6. Could authors give STD (i.e., gray area to represent STD) in Figure 4, 5 and 7? Some curves are too close, I am not sure they are statistically different.

Q7. How is the performance of the super-net?

Q8. Could the authors add an ablation study on this point? -  "The motivation is to improve all child models in our search space simultaneously, by pushing up both the performance lower bound (the smallest child model) and the performance upper bound (the biggest child model) across all child models."
- It is important to avoid fine-tune
- From the paper, I am not sure whether the problem is solved by changing the space or the proposed training method (See Q1).

**Experience Assessment:**

I have published one or two papers in this area.

**Review Assessment: Checking Correctness Of Derivations And Theory:**

I carefully checked the derivations and theory.

**Review Assessment: Checking Correctness Of Experiments:**

I assessed the sensibility of the experiments.

**Review Assessment: Thoroughness In Paper Reading:**

I read the paper at least twice and used my best judgement in assessing the paper.

---

> ### Author Response · Authors · 2019-11-13
> **Authors' Reply to Review**
>
> Thanks for your review efforts! We have addressed all questions below:
>
> General question: what’s the general lessons we can learn from the paper?
> In this work we demonstrated that it is feasible to train a single-stage model from which we can directly slice high-quality child models without retraining or finetuning. Our approach, in contrast to other weight sharing NAS methods, is simpler and has significant advantages in diverse deployment scenarios and resource-aware devices. To achieve this goal, we propose several simple and essential techniques backed up with extensive ablative study on large-scale ImageNet experiments.
>
> Q1: Thanks for the suggestion on the generalization of our method to other applications and datasets. We agree that applying the proposed single-stage architecture search method on other applications like detection and language tasks is an exciting direction. We hope our results on large-scale ImageNet can serve as a solid reference to these future work. Like many other related work including MobileNetV3, MNASNets and EfficientNets, our work mainly targets on searching efficient network architectures in large-scale ImageNet setting thus CIFAR and MNIST may not be meaningful.
>
> Q2: We mainly start from slimmable networks (Yu & Huang (2019b)) to study if it is possible to train a single-stage model that can instantly run with different architectural dimensions for scaling up neural architecture search. But the methods introduced in Yu & Huang (2019b) cannot deliver such high-quality single-stage models (for example, in Figure 2 the final performance of the biggest model is 1.6% worse than the best one). Then we identified that the failure is due to the distinct initialization and learning dynamics of small and big child models. We fixed it by proposing several simple and effective techniques and demonstrated the effectiveness with ablative studies.
>
> Q3: Our architectural search space is shown in Table 1. The coarse-to-fine selection (described in Sec 3.2 and Sec 4.4) is a simple and essential component in our NAS method to identify better child models under different resource budgets. Our work is novel as a simple, unified and effective approach to scale up neural architecture search, and the coarse-to-fine selection, compared with other search methods, is simple and effective and delivers good results.
>
> Q4: In our work, we claim that “We also use the swish activation (Ramachandran et al., 2017) and fixed AutoAugment V0 policy (Cubuk et al., 2018) following EfficientNets (Tan & Le, 2019).”, instead of "following other compared methods". This is because EfficientNet is the state-of-the-art and our strongest baseline. Recent mobile networks including MNasNet and MobileNet V3 used Swish function.
>
> Q5: Since during the search our approach does not require retraining or fine-tuning, the majority of our search cost (~99%) is in training a high-quality single-stage model. Our training cost is approximately three times more compared with training a normal network (e.g., training an EfficientNet). Moreover, by training a single-stage model, we are able to search models at different resource budgets, which saves the cost to scale up neural architecture search.
>
> Q6: Thanks for the suggestion. We will make our figures more clear in our final version. The curves in Figure 4, 5, 7 have differences of more than 0.5% Top-1 Accuracy on ImageNet, which are not close in the literature.
>
> Q7: The biggest child model (the performance of the super-net) has Top-1 accuracy of 81.8% with about 1.8 GFLOPs computation.
>
> Q8: In our work, we mainly generalize the Sandwich Rule (Yu & Huang, 2019b) to all architectural dimensions including resolution, width, depth and kernel size. The ablative study of the Sandwich Rule is shown in Table 2 in Yu & Huang, 2019b, which indicates that the Sandwich Rule is better than random sampling and other variations.

---

### Official Review · AnonReviewer2 · 2019-10-22
**Official Blind Review #2**

**Rating:** 3

**Review:**

The authors propose a search for neural architectures with different resource requirements by training a single model only. Furthermore, models found at the end of the search require no additional post-processing and are ready for deployment. A weight-sharing technique is applied to make this happen. The authors argue that there are multiple adaptions required to make it work. This includes the child model sampling strategy, use of model distillation, weight initialization, learning rate schedule, regularization and batch norm calibration.

The work seems to be an incremental extension of Yu & Huang (2019b) and phrased as a NAS algorithm. Many techniques considered vital for the proposed method make use of techniques proposed by Yu & Huang (2019b) (sandwich rule, inplace distillation, batch norm calibration and the way how weights are shared). Other required techniques are either proposed by others (initialization) or very simple (learning rate schedule and regularization). The authors claim that they extend the work by Yu &Huang (2019b) "to handle a much larger space where the architectural dimensions, including kernel sizes, channel numbers, input resolutions, network depths are jointly search". They never clarify why this is a non-trivial step and they might want to point this out in their rebuttal.

Besides this, the authors did a very good job. The paper is well-written, references are given wherever needed and all the closest related work is covered sufficiently. The experimental part conducts several ablation studies which supports their various decisions. Unfortunately, all results reported use heavy data augmentation which makes a comparison to other methods (besides EfficientNet) impossible. 600M is considered the upper bound for mobile architectures by the NAS community. Unfortunately, no such model is considered making it even harder to compare to existing NAS methods. The EfficientNet numbers reported don't match the ones reported in the original paper as far as I see. A red dot in Figure 3 could be added for the BigNASModel with 660M parameters.

**Experience Assessment:**

I have published in this field for several years.

**Review Assessment: Checking Correctness Of Derivations And Theory:**

I carefully checked the derivations and theory.

**Review Assessment: Checking Correctness Of Experiments:**

I carefully checked the experiments.

**Review Assessment: Thoroughness In Paper Reading:**

I read the paper thoroughly.

---

> ### Author Response · Authors · 2019-11-13
> **Authors' Reply to Review**
>
> Thanks for your review efforts! We have addressed all questions below:
>
> 1. Why is our work non-trivial?
> In this work we demonstrated that it is feasible to train a single-stage model from which we can directly slice high-quality child models without retraining or finetuning. Our approach, in contrast to other weight sharing NAS methods, is simpler and has significant advantages in diverse deployment scenarios and resource-aware devices.
>
> To achieve this goal, obtaining high-quality single-stage model is the key. However, naively following Yu & Huang (2019b) cannot deliver such high-quality big single-stage models that can instantly run with different architectural dimensions (for example, in Figure 2 the final performance of the biggest model is 1.6% worse than the best one, due to distinct learning dynamics of small and big child models). In this work, we proposed several simple and effective modifications and achieved better performance than all of our NAS baselines. We believe our proposed method and documented experiments are non-trivial in the literature and valuable to future neural architecture search methods.
>
> 2. The reported performance of EfficientNet doesn't match the ones reported in the original paper?
> We appreciate the reviewer’s effort on delving deep into our experimental details. We ensure experimental fairness by reporting the highest EfficientNet results based on the official published models [1], which are actually better than results in the original paper [2]. Our current experimental setting uses the same data augmentation as in the EfficientNet (our strongest baseline).
>
> 3. What’s the performance of BigNASModel at 600 MFLOPs instead of 660 MFLOPs?
> Although in our paper we only report a child model with 660 MFLOPs, one of our biggest advantages is that we can easily get child models around 600 MFLOPs (without retraining or finetuning) as requested: our child models have Top-1 accuracy 79.4% @ 545M; 79.5% @ 586M; and 79.6% @ 660M. We will update this result into our final version.
>
>
> [1] https://github.com/tensorflow/tpu/tree/master/models/official/efficientnet
> [2] Tan, Mingxing, and Quoc Le. "EfficientNet: Rethinking Model Scaling for Convolutional Neural Networks." International Conference on Machine Learning. 2019.

---

### Official Review · AnonReviewer3 · 2019-10-26
**Official Blind Review #3**

**Rating:** 6

**Review:**

This paper presents a method for architecture search in deep neural networks in order to identify scaled-down networks that can operate on resource limited hardware. The approach taken in this paper is different from other approaches, which train a single big model then fine tune smaller models for specific hardware, or use distillation to train progressively smaller models. Here, a single model is trained in a manner that allows subsequent slicing to smaller models without additional training required. The authors employ a variety of strategies to get this to work well, including specific initialization techniques, regularization methods, learning schedules, and a coarse-to-fine optimization method to obtain the smaller models. The authors demonstrate SotA performance on ImageNet relative to other techniques.

Overall, this was a well-written paper, and the results appear convincing. I would have liked a little bit more explanation about the implementation details though. As someone knowledgeable about deep ANNs, but not an expert in NAS for efficiency, I was not very clear on a couple of items. Specifically, I think it would be good to clarify the following:

1) The authors say that they use a masking strategy to implement slicing during training. So, do I take it that they effectively do a separate pass through the network with different masks to implement the different sized models? If so, do you then simply accumulate gradients across all masks to do the updates?

2) When the authors say they perform a fine grained grid search by varying the configurations, what is meant by this exactly? Do you mean that you do a search through different slicings of the big model to discover the best smaller architecture? What does it mean to do a grid search through binary masks? Or, is there a continuous hyperparameter that determines the masking which you do the grid search on? Maybe I’m just not understanding well at all, but even still, in that case, it indicates that this could be clarified in the manuscript.

**Experience Assessment:**

I do not know much about this area.

**Review Assessment: Checking Correctness Of Derivations And Theory:**

N/A

**Review Assessment: Checking Correctness Of Experiments:**

I assessed the sensibility of the experiments.

**Review Assessment: Thoroughness In Paper Reading:**

I read the paper at least twice and used my best judgement in assessing the paper.

---

> ### Author Response · Authors · 2019-11-13
> **Authors' Reply to Review**
>
> Thanks for your review efforts! We have addressed all questions below:
>
> 1. Yes we perform several “separate passes” through the network with different masks to implement slicing different sized models during training. We simply accumulate the gradients of the weights that are not masked, across all “separate passes” (e.g., we sample 4 child models according to the sandwich rule in each training iteration). Put it in the pseudo code, in each training iteration we perform:
>
> > for NetConfig in SampledConfigList:
> >        run child model of NetConfig;
> >        compute loss and store gradients temporarily (only on the weights that are not masked);
> > end for
> > accumulate all pre-computed gradients;
> > apply gradients to the weights;
>
> 2. By varying the configurations in a fine-grained grid search, we aim to find better child models. It is achieved by randomly mutating its network-wise resolution, stage-wise depth, number of channels and kernel sizes. The search is on a discrete space where, for example, the resolution can be 208, 224, 240, the number of channels can be 32, 48, 64, etc. The search is performed by loading the partial weights of a particular configuration and running the actual forward inference (we use binary masking only during the training, during search and inference we use the exact network architecture).

---

### Decision · Program_Chairs · 2019-12-19

**Decision:**

Reject

**Comment:**

This paper presents a NAS method that avoids having to retrain models from scratch and targets a range of model sizes at once. The work builds on Yu & Huang (2019) and studies a combination of many different techniques.
Several baselines use a weaker training method, and no code is made available, raising doubts concerning reproducibility.

The reviewers asked various questions, but for several of these questions (e.g., running experiments on MNIST and CIFAR) the authors did not answer satisfactorily. Therefore, the reviewer asking these questions also refuses to change his/her rating.

Overall, as AnonReviewer #1 points out, the paper is very empirical. This is not necessarily a bad thing if the experiments yield a lot of insight, but this insight also appears limited. Therefore, I agree with the reviewers and recommend rejection.